# LIFELONG CONTROL THROUGH NEUROEVOLUTION

## ABSTRACT

Reinforcement learning (RL) under continual environmental changes has remained a central challenge for decades. Novel designs of loss functions, training procedures and neural network architectures have not yet managed to alleviate the main mode of failure in lifelong learning: loss of plasticity. Here, we turn to a very different family of optimisers: neuroevolution (NE). Through an extensive evaluation on diverse lifelong control tasks, we see that NE exhibit a remarkable ability to adapt where RL fails catastrophically. We observe that, in the present of environmental shifts, NE naturally increases its diversity of solutions, evolving the ability to rapidly discover well-performing specialist individuals. We propose that NE can be a promising approach towards tackling the need for lifelong adaptation and that future work in both RL and NE should focus on the benefit of diversity in population-based approaches.

## 1    INTRODUCTION

Reinforcement learning (RL) has produced impressive feats in recent years, offering ways to advance robotic agents (Silver et al., 2016b), improve the capabilities of Large Language Models (Ouyang et al., 2022) and perform on-par with humans in complex decision-making problems (Silver et al., 2016a). It is not surprising that RL, the de facto formalism for learning through interaction with an environment, is driving the increasing adoption of Artificial Intelligence (AI) in the real world. However, today's RL algorithms require immense training experience, extensive hyper-parameter tuning and, nevertheless, remain brittle to unexpected variations in their environment (Kirk et al., 2023; Pan et al., 2025; Kudithipudi et al., 2022). Continuing on this path requires moving out of our current paradigm of close human supervision and towards AI that can autonomously learn in shifting, *open-ended* worlds (Hughes et al., 2024; Clune, 2020). A major mode of failure for RL in such settings is a *loss of plasticity* (Klein et al., 2024; Sokar et al., 2023; Muppidi et al., 2024). Plasticity lies at the other end of stability and, since a large focus of past progress, RL has been on improving stability (Schulman et al., 2017b; Kirkpatrick et al., 2017), RL's best-performing algorithm often exhibits a remarkable tendency to ignore shifts in their environemnt (Klein et al., 2024). The recent resurgence of RL has, however, triggered an interest in bringing plasticity back. A natural approach is the development of techniques that re-trigger learning (Sokar et al., 2023). However, such approaches often require extensive tuning, defeating their original motivation (Muppidi et al., 2024; Klein et al., 2024). Lifelong RL is today in need of low-cost, hyperparameter-free solutions for battling loss of plasticity. Here we propose to look for such a solution in a fundamentally different family of optimisers: neuro-evolution.

Evolutionary algorithms are a long-standing and diverse family of black-box optimization methods that, when applied for the optimization of artificial neural networks (ANNs), are referred to as neuroevolution (Risi et al., 2025; Stanley et al., 2019). A core component of any evolutionary algorithm is the presence of a *population*. The field originally drew its inspiration from biological evolution, repurposing the processes of mutation, selection, and reproduction for black-box optimization (Koza). Since then, the field has grown into an engineering-focused discipline with a pluralism of methods that leverage the population in diverse ways Risi et al. (2025). NE methods have recently been shown to perform competitively with RL in control tasks, with previous works noting that their population-based nature brings several advantages: scalability through parallelization, increased exploration that helps avoid local optima thanks to population diversity, and smaller sensitivity to hyperparameters (Such et al., 2018; Salimans et al., 2017; Chalumeau et al., 2023). In this work, we propose that there is another, yet unexamined, benefit of NE in control tasks: avoiding loss of plasticity in the face of environmental shifts.

Environmental change is both a setback and a driver for evolution. In a constant environment and in the absence of an explicit mechanism for preserving diversity, a population converges to minimal diversity imposed by its mutation rate (Giraud et al., 2001). When a shift occurs, it causes increased fitness variation in the population, retriggering competition amongst the population and, thus, exploration. If the shift is too large or the population is not sufficiently diverse, the population may experience a mass extinction (Nisioti & Moulin-Frier, 2022). If, however, some individuals survive, this increased diversity can act as a buffer from future environmental changes. Thus, the population becomes increasingly better at dealing with variation. The form of environmental change plays an important role in this with some types of variation favoring adaptability (Nisioti & Moulin-Frier, 2022; Grove, 2014). The idea is central in the field of artificial open-endedness (Soros, 2017), where evolving environments alongside agents trigger an automated curriculum driving continual change (Clune, 2020; Wang et al., 2019). Works in this community have, in particular, shown that evolution can handle and often benefits from environmental variability (Lehman & Miikkulainen, 2015).

Despite its relevance, to the best of our knowledge, no previous study has attempted to benchmark evolution's abilities in lifelong learning against those of reinforcement learning. Here, we offer such a study. We consider a collection of tasks that pose a wide diversity of challenges, such as sparse exploration and control of large ANNs, including feedforward, convolutional networks and a Transformer-based architecture. We examine two distinct NE approaches: genetic algorithms (GA) and evolution strategies (ES). We benchmark them against PPO (Schulman et al., 2017a), a state-of-the-art RL algorithm as well as a recent variant explicitly designed for such lifelong settings. Our study shows that NE exhibits an impressive ability at learning in the phase of environmental shifts, surpassing the state-of-the-art in lifelong control.

To understand how NE achieves this, we analyze the population dynamics in the presence of environmental shifts. In particular, we focus on the diversity within the population. We observe that, under environmental shifts, populations evolve higher diversity compared to those in a constant environment. In a long evolutionary run, we see that, after prolonged environmental variability, the population experiences a phase transition: a gradual increase in diversity leads the population towards an abrupt shift to the optimal solution This ability is contingent on the size of the population, with a small population instead experiencing a collapse. Isolating the highest performing individuals in a given shift reveals they differ significantly from each other: evolution adapts by finding specialist agents rather than a single generalist agent. ES differs significantly in how it deals with variation in comparison to the GA: diversity remains low, and the population progresses slowly. Overall, the GA outperforms ES, except when faced with sparse reward problems.

We provide code for reproducing our study at an anonymous github repo.

## 2 BACKGROUND AND RELATED WORKS

The problem of *plasticity loss* in artificial neural networks (ANNs) has been recognized since the 1980s, when catastrophic forgetting was identified as a key weakness of gradient-based learning compared to symbolic or non-connectionist approaches (McCloskey & Cohen, 1989). In deep reinforcement learning (RL), where ANNs serve as policy approximators, this issue is magnified by the non-stationarity of the environment. Algorithms such as PPO introduced stabilizing mechanisms like clipped objectives and trust regions (Schulman et al., 2017a), which improved reliability but further reduced adaptability. Continual RL methods have attempted to restore plasticity through techniques like dynamic regularization or masked networks (Muppidi et al., 2024), but they often require privileged information about when shifts occur and remain brittle under rapid change. Even large-scale pre-trained models such as LLMs exhibit similar fragility when deployed out of distribution (Kirk et al., 2023), suggesting that the stability–plasticity dilemma persists across domains.

Population-based methods, such as neuroevolution (NE), provide a fundamentally different way to address this challenge. Instead of optimizing a single solution, they maintain a diverse set of candidates that evolve over time. Diversity allows populations to explore multiple adaptive pathways, recover from dead ends, and naturally specialize after environmental shifts (Salimans et al., 2017; Chalumeau et al., 2023). This makes NE well-suited for lifelong learning, where continuous adaptation is essential. Related approaches in multi-task learning, such as mixtures of experts or modular networks, also leverage specialization, but they rely on a fixed set of experts and explicit coordination mechanisms, making them less open-ended than evolutionary search.

In this work, we compare two representative evolutionary algorithms. The *genetic algorithm* (GA) explicitly maintains a population, selecting top-performing individuals and generating new candidates through mutation while preserving an elite subset for stability (Such et al., 2018). *OpenES* (Salimans et al., 2017) provides a scalable alternative by maintaining only the mean and sampling perturbations, sacrificing rich modeling for efficiency.

# 3 EMPIRICAL STUDY

How does NE behave in control tasks with shifting dynamics? Does it exhibit similar limitations to RL, in particular when it comes to loss of plasticity? To answer these questions, we here evaluate a variety of NE and RL algorithms in lifelong learning tasks with the aim of observing and understanding differences in their ability to handle environmental shifts. We focus on two distinct NE approaches: a genetic algorithm (GA) (Such et al., 2018) and an evolution strategies (ES) (Salimans et al., 2017). We benchmark these methods against PPO, a state-of-the-art RL algorithm, and Trac-PPO, a variant that adds adaptive regularization to PPO to address the loss of plasticity (Muppidi et al., 2024) [1].

To ensure a fair comparison, we employ the same budget of experience for the different approaches. We have matched the sample complexity of the methods by considering a fixed number of episodes for NE in a certain environment and multiplying this number by the population size and episode length to determine the number of steps for PPO. While it is possible to match complexity through other metrics, such as execution time, we believe that, from the perspective of lifelong learning in the real world, environment steps are what matters. All conditions have been run for 10 trials, and we provide mean and variance estimates.

**Note on hyperparameter tuning**  : We have tuned all methods separately for each family of tasks (but not for each task; we picked a random task within a family) to ensure that the methods work well for the classical version of all tasks. In some cases, we have employed well-performing hyperparameters suggested in previous works. We have not performed exhaustive tuning for NE: we have employed the default hyperparameters provided by evosax (Lange, 2022a) and manually searched for well-performing values when needed. While we expect benefits in some conditions by further tuning, we believe that it should not play a central part in our study. First, a major reason for the attractiveness of NE approaches is their remarkable robustness to hyperparameters compared to other deep learning algorithms(Chalumeau et al., 2023; Such et al., 2018). Second, when comparing NE approaches, our focus is not on their performance but on their distinct dynamics, which, due to the aforementioned robustness, do not depend on the hyper-parameterization except for edge cases.

We first, describe the tasks we have considered in Section 3.1. We, then, take an overall look at performance differences. across all tasks and methods in Section 3.2. In Section 3.3, we dive into an analysis of the behavior of NE.

Our study is accompanied by appendices that provide implementation details and hyperparameters employed, and additional empirical results that we refer to throughout the rest of the paper.

## 3.1 TASKS

We consider three task families for lifelong learning, which we explain next.

**Classic control**   We use the tasks Cartpole, Acrobot, and MountainCar, implemented in the gymnax library (Lange, 2022b). These are test-beds with discrete actions and continuous observations that pose simple control challenges and can be solved by small feedfoward networks. Cartpole is an easy balancing task with immediate and dense rewards. Acrobot and MountainCar are more challenging sparse reward environments, with the latter being significantly easier as good solutions can be reached through random actions.

For each of the three tasks, we employ a lifelong-learning variation originally introduced in Muppidi et al. (2024). Every 200 generations, we sample a vector from a normal distribution with standard deviation 1.0 and add it element-wise to the observations (i.e., the vector remains constant for 200

---

[1]We currently provide the performance of Trac-PPO only in the simple control environments. We have a

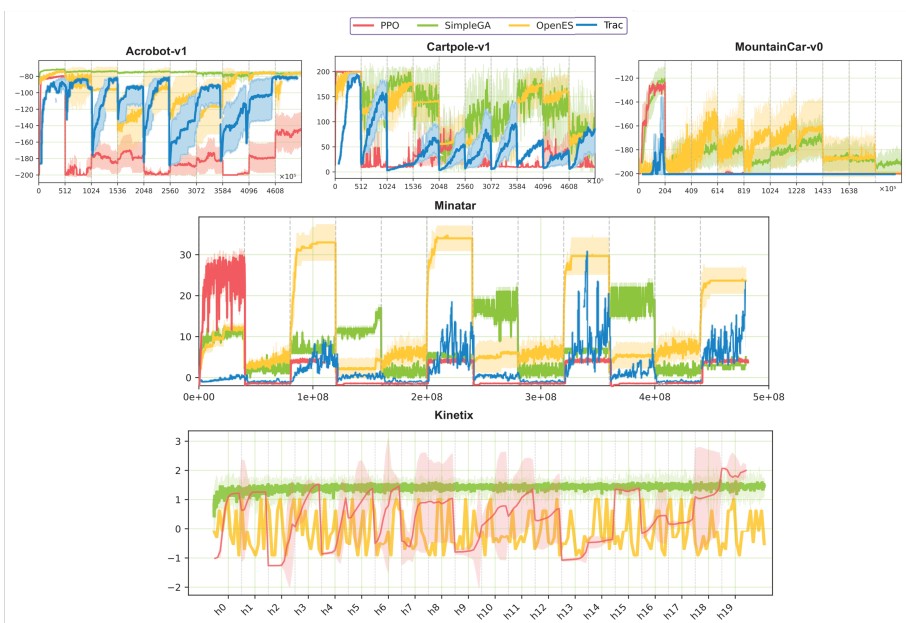

Figure 1: **Overall comparison across tasks**: we benchmark two NE approaches (SimpleGA and OpenES) against two RL variants (PPO and Trac-PPO, a variant designed specifically for lifelong learning). (Top) Simple control tasks (Middle) Minatar tasks: we consider the order Breakout, Asterix, Space-Invaders, going through four phases. (Bottom) Kinetix enviroments of medium difficult.

generations). This variation was found to be more challenging for RL compared to other approaches that vary the dynamics of these tasks. It can be seen as a model of distribution shifts in the sensory perception of the agent.

**Minatar** From the popular Atari benchmark we use Breakout, Asterix, SpaceInvaders, all implemented in the gymnax library (Lange, 2022b). Originally introduced by Young & Tian (2019), these tasks simplify the original Atari games by reducing the size of the grid and replacing RGB observations with pixel-centric symbolic information. Due to their lower complexity, they are often solved with feedforward networks by collapsing the input pixels. Here, we employ CNNs to extend the breadth of architectures considered in our study. Achieving a high score in these games requires sophisticated strategies to deal with complex challenges, such as exploration under high risk in Asterix, temporal credit assignment in Breakout and long-term planning in SpaceInvaders. Despite this reduction in complexity, these tasks still pose challenges for NE (Lange et al., 2023) and some RL methods.

As commonly done in continual learning studies, we chain these environments one after another in repeated phases with the order: Breakout, Asterix, SpaceInvaders. (Muppidi et al., 2024).

**Kinetix** Kinetix is a test-bed for testing the generalisation capabilities of RL agents (Matthews et al., 2024). It contains a variety of procedurally-generated and handcrafted environments that vary in their complexity and size. A Transformer-based architecture enables controlling robots of varying morphology and solving tasks differing in their input/output size. The ANN employed here is significantly larger than the networks we used in the previous tasks (about 750000 parameters). To the best of our knowledge, our study is the first to attempt optimizing a Transformer of this size with NE. As we will see, the GA exhibited impressive performance in these tasks while for ES we did not manage to find a well-working solution (even without shifts, see our note on tuning above).

To formulate a lifelong learning set-up, we employ the set of 20 manually designed tasks of medium difficulty and go through them in a sequence. While they have not been specifically designed to exhibit a curriculum, these tasks become increasingly progressive.

We refer to intervals between shifts in the environments as phases. When we employ the original version of an environment, without environment shifts, we refer to is as the original version.

## 3.2 OVERALL COMPARISON

Figure 1 presents an overall comparison for the different methods and tasks. In particular, we visualize the progression of rewards accumulated in a given episode across training, where the horizontal axis indicates the number of environmental steps passed for a method. Shifts in the environment are indicated with vertical dotted lines. For the RL methods, these values are computed by evaluating the current policy in deterministic episodes, where we average across 20 seeds for the environment. For the evolutionary approaches, the fitness of the best individual in the current generation/episode is reported (which is an average across 20 environment seeds). This process is repeated 10 time to get the reported means and confidence intervals. These results are accompanied by a table with cumulative fitness scores and tests for statistical significance in Figure 7.

To facilitate our analysis, we also provide in Figure 5 of Appendix B the performance of methods in the absence of environmental shifts. This information is necessary, as low performance in the continual learning setting may be due to the inability of a method to master the task rather than the added challenge of shifts.

**Overall failure of RL**  We observe that methods exhibit significant differences in their performance. PPO succeeds in the first phase but fails in subsequent ones, being the lowest performing method in most tasks (the only exception is OpenES in Kinetix, but as we will see later, OpenES could not master this task ). As we see in Figure 5, PPO is the best-performing method in the normal versions of these tasks, so its failure is caused by the shifts. The failure of PPO is particularly pronounced in the simple control tasks, where it performs worse than random search (Acrobot and Cartpole can be easily solved through random search (Oller et al., 2020)). In Minatar, PPO converges to complete failure in two of the tasks, while accumulating a small reward in one of them (Breakout). This result indicates a complete loss of plasticity. In the Kinetix environments, PPO solves some of the tasks (9 out of 20) in some trials, exhibiting high instability. In contrast, in the original version of these tasks (Figure 5), PPO solves 19/20 tasks. Trac-PPO improves upon the performance of PPO in all cases, but its performance remains unstable and lower than the one achieved by the NE methods.

Turning to the NE methods, we observe that they both accumulate high rewards during environmental shifts, performing comparably to each other. The relative performance of these methods varies in the original version of these tasks, so to compare them, we need to carefully examine each condition in isolation.

**NE in lifelong classic control**  We first turn to Acrobot, where we observe that both methods perform well, with the GA exhibiting impressively steady good performance. In the absence of shifts, both methods solve the task (this is true for all three tasks in this family) but ES converges significantly more slowly, requiring about 200 generations. This is arguably the reason for its lower performance in the lifelong setting: as we see in Figure 1 ES improves within each phase and solves some phases perfectly. [2] In Cartpole, methods perform comparably and do not achieve a full recovery (we will look more closely into this task and see how the GA can master it under long-term evolution in Section 3.3). In MountainCar, we observe that OpenES outperforms the GA to some extent. This is particularly interesting as, in the original version of this task, GA converges to the optimal solution much quicker. This suggests that the GA is more challenged than ES when facing environment shifts in this task. We will look deeper into this behavior in the next section

---

[2]Here we should note an important feature we noticed in our experiments: adding a vector to the observations of the environment can make a task more difficult for NE even in the absence of shifts. This is arguably due to the change in the range of the observations NE methods received. Differently from RL, NE here and commonly does not employ techniques for normalizing its inputs. This means that depending on its initialization it can start off in a bad region. Therefore, we should note that the performance reported for the original version of these tasks is an upper bound rather than an expected value for the performance in phases during lifelong learning.

**NE in lifelong Minatar**  While NE methods were significantly outperformed by RL in the classical version of these tasks, we see that in the continual version, both NE methods perform significantly better. In particular, the GA exhibits the performance in the first task (Breakout) while ES exhibits the best performance in the third task (SpaceInvaders). With the exception of a large drop in the performance of the GA for SpaceInvaders, NE has remained largely unaffected (yet low-performing) by the shifts In the next section, we will closer into this failure of the GA and relate it to its failure in MountainCar

**NE in Kinetix**  ES did not manage to solve this task, but the GA exhibited impressive performance. We observed that:

- For the small tasks, PPO-Transformer can find the optimal solution in both the normal and lifelong set-up. Interestingly, the last two tasks are only solved in the lifelong set-up, which means that the agent benefits from being pre-trained. Thus, lifelong learning is not an issue here.

- The large tasks cannot be solved in the normal setup. It is likely that these tasks are too difficult to solve without pre-training.

- For the medium tasks, we observe that the normal setup works well, but under lifelong learning, performance degrades: the tasks are solved either much more slowly or are never solved.

We use the manually designed tasks of medium size

## 3.3 Insights into evolution

Having noted multiple intriguing behaviors in our overall comparison, we turn towards a more in-depth look into the dynamics of NE populations. Considering the surprising adaptability of NE, we would like to develop an intuition on how populations react and manage to deal with shifts. We are equally interested in understanding what happens when they fail to do so.

Understanding why a method optimising a policy parameterised by a neural network fails or succeeds in a control task is not an easy feat. The complexity of environmental dynamics, the black-box nature of ANNs, and our frequent lack of a formal analysis of the search method stand in the way. Large effort in the supervised and reinforcement learning community is nevertheless put into developing analysis techniques with many notable successes (Sokar et al., 2023).

Due to their population-based nature, NE approaches often lay emphasis on properties such as the diversity of the population. For some of the most popular approaches, such as Quality-Diversity (Chalumeau et al., 2023) , diversity is not just an afterthought but an explicit optimization objective or constraint. When it comes to methods solely optimizing for performance, studies rarely go into a post-hoc analysis. Exceptions to this primarily come from work in evolutionary optimization that employed small search spaces (Grove, 2014; Ouyang et al., 2022). Applying such analysis in the search spaces considered in modern NE significantly increases the memory and computational requirements of a study. Thus, unsurprisingly and to the best of our knowledge, studies of diversity in NE remain underdeveloped.

Here we turn to such a study in the classic control tasks we consider. We measure the diversity in a population as the mean

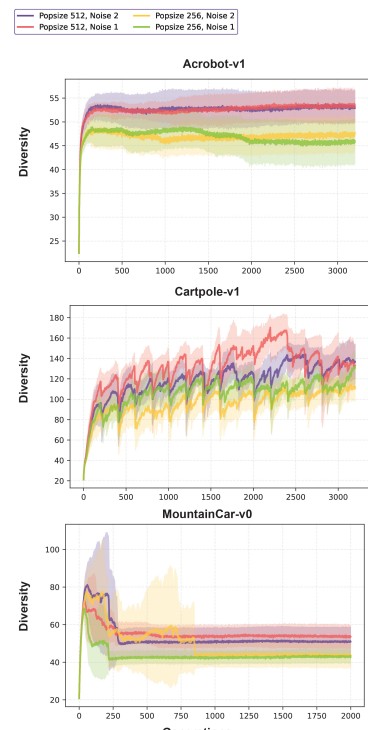

Figure 2: **Population diversity in the classic control tasks**. We study how diversity varies for two different population sizes and two levels of noise.

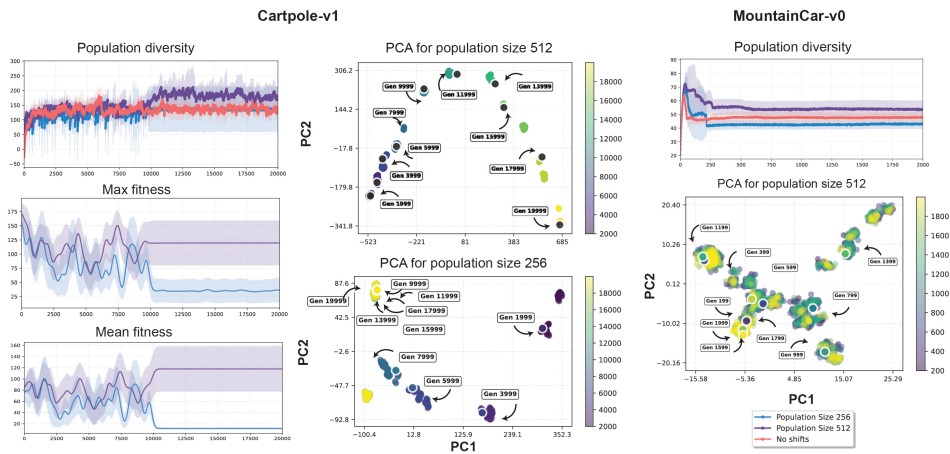

Figure 3: A diversity analysis of GA on (left) Cartpole, and (Right) Mountaincar.

pairwise distance computed in the original parameter space. In addition, we employ a dimensionality reduction technique (PCA) to visualize the trajectories that the population follows across evolution.

**Shifts promote diversity**  In Figure 1 we observed that the GA remains robust to shifts in Acrobot but exhibits some instability in Cartpole. We found this intriguing as the two tasks exhibit similar dynamics and complexity. Upon monitoring how diversity evolves for these tasks (Figure 2) we see that the diversity in Acrobot stabilised early on, while it was still ascending for CartPole at the end of this experiment. This observation motivated us to run a very long experiment, simulating 100 environmental shifts. To our surprise, the population exhibited a stark transition around 900 generations (see Figure 3). Whether the transition was a favorable outcome or not depended on the population. For a large population (512 individuals) 8 out of 10 trials optimally solved the task, and the rest converged to a relatively high value. When the population was smaller (256 individuals), 9 out of 10 trials converged to a minimal fitness. Looking at the diversity in this longer experiment, we see that, right at this transition point, the large population abruptly increases its fitness. Overall, this analysis shows that diversity is driven by non-stationarity in the environment and that, in its turn, drives the ability of populations to avoid a mass extinction. Different from the GA, OpenES does not adapt by increasing its diversity but by adapting gradually (see Figure 6). This is not surprising as OpeES is a distribution-based approach that does not carry over the population across generations but spawns it, assuming a Gaussian distribution around its learned mean.

**Diversity is not sufficient**  We now turn towards the MountainCar, one of the cases where the GA exhibits a lower ability at handling non-stationary. As we see on the bottom of Figure 2 and on the right of Figure 3, diversity here exhibits a form qualitatively different from the other two tasks. It increases initially (up to encountering the environment shift and then drops randomly. When looking at the PCA plots for this task, we see a starkly different behavior: there is no clear progression with generations. Our hypothesis is that, due to the sparse reward nature of this task, the population does not exhibit sufficient variance in its fitness for fitness-based selection to offer an improvement. When looking at the mean performance of the population in this task, we observe that all agents are failing to collect rewards.

## 4  DISCUSSION

Our study demonstrates that neuroevolution (NE) offers a compelling alternative to reinforcement learning (RL) for lifelong learning in dynamic environments. While RL methods such as PPO excel in stationary settings, they suffer from severe loss of plasticity when faced with environmental shifts, often converging to complete failure modes. In contrast, population-based NE approaches adapt naturally to change by maintaining and exploiting diversity within the population. This adaptability enables NE to discover specialist solutions after each shift rather than relying on a single, increasingly rigid generalist policy. These findings position NE not as a replacement for RL, but as

a complementary paradigm for situations where continual adaptation and long-term autonomy are essential.

While NE proved remarkably robust, our analysis also revealed its limitations. In particular, the genetic algorithm (GA) struggled in sparse-reward tasks such as MountainCar and certain Minatar environments, where diversity alone was insufficient to guide evolution toward higher-fitness solutions. Similarly, the evolution strategies (ES) method showed slower adaptation overall and failed to handle high-dimensional architectures like the Transformer-based controller in Kinetix. Furthermore, our diversity analysis suggests that population size plays a critical role: larger populations can undergo phase transitions leading to successful adaptation, whereas smaller populations risk collapse under rapid environmental change.

Looking ahead, our results open several exciting research directions. Future work could explore the co-evolution of mutation rates, network architectures, and even environmental complexity to further improve plasticity and scalability.

We believe that our observation that increased diversity can be useful in non-stationary environments can offer insights that span beyond the field of NE and into RL. In particular, distributed RL has shown promising results not just in helping scale up RL but also improving upon its performance, with solutions hypothesizing that the benefits come from increased diversity (Horgan et al., 2018) Investigating how NE can operate alongside RL in hybrid frameworks may yield agents that balance the stability of gradient-based learning with the adaptability of evolutionary search. Finally, expanding benchmarks to more open-ended environments and real-world robotics tasks will help clarify the limits of NE in practical applications. By demonstrating that populations can overcome the plasticity-stability dilemma without extensive tuning or external supervision, this work suggests that evolution remains a powerful and underutilized tool for building lifelong learning systems.

Note: We have used LLMs solely for polishing text.

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
