 

Figure 4: First two kinetix tasks: (left) training both tasks from scratch works; (right) training first on task h0 and then warm-starting h1 with the solution of h0 fails.

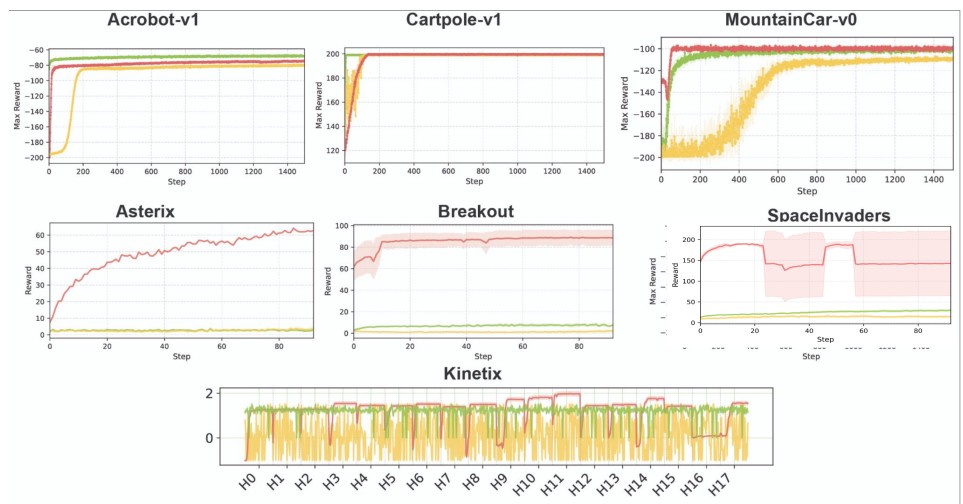

Figure 5: **Performance without shifts in tasks** here we just show that all methods work when there are no shifts, some need longer than others

Kenny Young and Tian Tian. Minatar: An atari-inspired testbed for thorough and reproducible reinforcement learning experiments. *arXiv preprint arXiv:1903.03176*, 2019.

## A  TEMPORARY RESULTS FOR REBUTTAL

## B  OVERALL COMPARISON: ADDITIONAL RESULTS

## C  METHODS AND HYPERPARAMETERS

This section contains more details about the experimental set-up, including the hyperparameters employed.

### C.0.1  CLASSIC CONTROL TASKS

We consider 3 classic control tasks, Acrobot-v1, Cartpole-v1 and MountainCar, implemented in the gymnax library Lange (2022b). The episode lengths are 200, 200 and 500, respectively. The optimal reward for Acrobot is -60, for Cartpole 200, and for MouuntainCar -100. We employ the same MLP

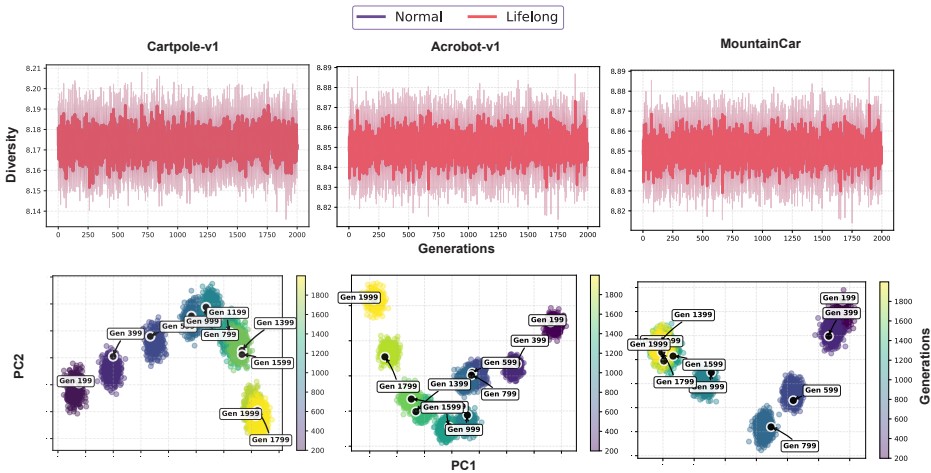

Figure 6: **Cumulative fitness score and statistical test for significance for the classic control environments**

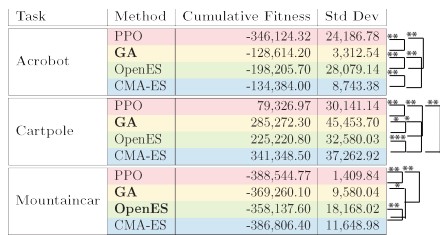

| Task | Method | Cumulative Fitness | Std Dev | |
|------|--------|-------------------|---------|---|
| Acrobot | PPO | -346,124.32 | 24,186.78 | ** ** |
| | **GA** | -128,614.20 | 3,312.54 | ** ** |
| | OpenES | -198,205.70 | 28,079.14 | ** |
| | CMA-ES | -134,384.00 | 8,743.38 | |
| Cartpole | PPO | 79,326.97 | 30,141.14 | ** ** ** |
| | **GA** | 285,272.30 | 45,453.70 | * * |
| | OpenES | 225,220.80 | 32,580.03 | *** |
| | CMA-ES | 341,348.50 | 37,262.92 | |
| Mountaincar | PPO | -388,544.77 | 1,409.84 | ** ** |
| | **GA** | -369,260.10 | 9,580.04 | |
| | **OpenES** | -358,137.60 | 18,168.02 | ** |
| | CMA-ES | -386,806.40 | 11,648.98 | |

Figure 7: **Diversity of OpenES in classic control tasks**

architecture for the three tasks: 2 hidden layers of 16 neurons and a bias neuron, ReLU activation functions at hidden and a linear activation function at action neurons.

For each task, we implement a continual-learning version in the following way: every 200 generations, we sample a new vector from a normal distribution with standard deviation 1.0 and add it element-wise to the observations. This vector remains constant for 200 generations. This variation of the classic control tasks was first proposed by (Muppidi et al., 2024), where it was found to be more challenging than other variations that change the environment dynamics. It can be seen as a model of distribution shifts in the sensory perception of the agent.

We present the hyperparameters on our online repo.

### C.0.2 KINETIX

We use the same Transformer architecture as the one employed by Matthews et al. (2025). Kinetix offers manually-designed tasks of three sizes: small, medium, and large. To implement the lifelong learning set-up, we pick one family of tasks and then sequentially move through them (we warm-start training in the new task using the parameters and optimizer state of the previous task).

We observed that:

- For the small tasks, PPO-Transformer can find the optimal solution in both the normal and lifelong set-up. Interestingly, the last two tasks are only solved in the lifelong set-up, which means that the agent benefits from being pre-trained. Thus, lifelong learning is not an issue here.

- The large tasks cannot be solved in the normal setup. It is likely that these tasks are too difficult to solve without pre-training.

- For the medium tasks, we observe that the normal setup works well but under lifelong learning, performance degrades: the tasks are solved either much more slowly or are never solved.

We use the manually designed tasks of medium size