# OpenReview forum: "Lifelong control through Neuro-Evolution"
_ICLR.cc/2026/Conference — Submitted to ICLR 2026_

### Official Review · Reviewer_FkgN · 2025-10-17

**Soundness:** 2
**Presentation:** 1
**Contribution:** 3
**Rating:** 4
**Confidence:** 4

**Summary:**

The authors show that neuroevolution methods perform better than deep RL methods in the context of environment shifts as they seem to suffer less from a loss of plasticity. They compare a GA and OpenAI ES against PPO and TRAC-PPO on a variety of environments: 3 classic control environments, 3 minatar environments and medium difficulty Kinetix environment.

**Strengths:**

The question addressed by the authors is interesting, the methodology looks sound and the results are of interest.

**Weaknesses:**

The paper suffers from severe editing issues:
- it refers to appendices, but appendices are absent. In particular, the paper refers a lot to Fig. 6, but we cannot see it. It is hard to provide a thorough review in this context...
- Figure 1 is supposed to show the performance of GA, OpenAI ES, PPO and TRAC-PPO, but TRAC-PPO is not shown
- the paper mentions an anonymous github repo but the url is only available in the "pdf-with-hyperlink" format. It happens that this github repository violates anonymity rules.
- the paper is full of typos.

Beyond that, there are scientific issues:

- the authors claim that failure of RL in the context of environment shifts is due to a loss of plasticity (or they consider that this failure is by definition a loss of plasticity), but the mechanisms behind these failures are not investigated in the light of the plasticity loss literature.

**Questions:**

Footnote 1 is unclear: what vector do the authors add? Why?

In Kinetix, the authors "use the manually designed tasks of medium size(.)". Can they be more specific about these tasks (what do they consist of, how many are them, etc.)?

Could you expand the caption of Fig. 3, to better explain what should be seen?

Typos:
- the authors are often using the future when they should use the present. Track "will" are remove the ones that do not seem necessary.
- abstract: "in the present of" -> presence
- the Koza citation is inadequate. Please provide all the necessary fields (year, etc.)
- there are two references for the same Michael Matthews' paper. Please merge them.
- Risi et al. (2025) -> use \citep{}
- p1: avoidng
- p2: distnct
- solution This -> missing dot
- it deal(s) with
- p3, line 112: missing ref (?)
- algorithms(Chalumeau -> missing space
- p4: pose pose
- p5: 10 time -> times
- We, first, turn -> rephrase and give more context
- argueably -> arguably
- p6: th performance -> the
- shifts In  -> missing dot
- medium size  -> missing dot
- optimiszation -> choose s or z :)
- p7: stationartiy ... the ablity ... exhibitnt ... thebottom
- p8: environements -> environments

---

> ### Author Response · Authors · 2025-11-18
> **Overall reply to reviewer's comments**
>
> We thank reviewer FkgN for their comments, we found them all useful and will explain below the changes they have lead to/will lead to. We are currently revising the paper (we will post a general reply detailing all changes soon). We also thank the reviewer for letting us know about their availability and hope they can still view this.
>
> **Comment** It refers to appendices, but appendices are absent. In particular, the paper refers a lot to Fig. 6, but we cannot see it. It is hard to provide a thorough review in this context.
>
> We apologise for the omission; we erroneously thought we had provided the appendix. We have just uploaded a version that we will continue improving until the deadline and notify you of any changes.
>
> **Comment 2** Figure 1 is supposed to show the performance of GA, OpenAI ES, PPO and TRAC-PPO, but TRAC-PPO is not shown.
>
> Indeed, we had accidentally provided an older version of that figure. In the revised manuscript we present the results for TRAC-PPO for the simple control tasks and Minatar. We also ran TRAC-PPO on kinetix tasks and will have those results in a couple of hours. For now we provide the results for the two first tasks of kinetix in the Appendix, Figure 4 where we can see that there is a drop of performance already with the first transition
>
> **Comment 3**. the paper mentions an anonymous github repo but the url is only available in the "pdf-with-hyperlink" format. It happens that this github repository violates anonymity rules.
>
> We have double-checked the anonymous repo link we have provided and could not find any violation of anonymity. Was this comment perhaps by mistake?
>
> **Comment 4**. the paper is full of typos.
>
> We apologise for the typos. We  have already addressed most of the typos and will do a thorough rewrite of the text once we have the results we are preparing for the rebuttal.
>
> **Comment 5** the authors claim that failure of RL in the context of environment shifts is due to a loss of plasticity (or they consider that this failure is by definition a loss of plasticity), but the mechanisms behind these failures are not investigated in the light of the plasticity loss literature.
>
> We find this observation particularly useful, thank you. We are currently preparing an analysis of the loss of plasticity of RL, where we will:
>
> a) present the size of gradient updates for PPO and observe that they become very low after convergence
> b) present the number of dormant neurons (see [Original paper](https://proceedings.mlr.press/v202/sokar23a/sokar23a.pdf)) and observe that they increase for PPO
>
> In relation to this we will also:
>
> a) measure the backward/forward transfer for all methods (this is the number of environmental interactions required to solve the next/previous task). Low forward transfer and high backward transfer indicates plasticity loss
>
> b) present an additional baseline and perform the same analysis on it (the DQN version designed to deal with dormant neurons).
>
> **Comment 6**. Footnote 1 is unclear: what vector do the authors add? Why?
>
> We understand the confusion of the reviewer, as this vector was described quite a few pages before (the last line on Figure 3 of the original and revised paper). This is the vector added to the observations in the simple control tasks.
>
> **Comment 7** In Kinetix, the authors "use the manually designed tasks of medium size(.)". Can they be more specific about these tasks (what do they consist of, how many are them, etc.)?
>
> We agree with the reviewer that this was poorly explained. We will add the following text in the revised manuscript:
>
> Kinetix contains a diversity of manually-defined tasks that differ in their difficulty. These tasks are categorised as small, medium and large, with larger environments containing more of the basic building blocks of the engine (eg circles, squares, actuators and motoris). It is assumed that tasks with more items are more complex and, therefore, more difficult. Kinetix contains 20 medium-size tasks.
>
> We will also provide some images and a more detailed discussion of this engine in the appendix.
>
> **Comment 8**. Could you expand the caption of Fig. 3, to better explain what should be seen?
>
> We agree with the reviewer that the previous caption was very short/missing important information. As we will explain in our overall reply to reviewers, we are preparing a better analysis of diversity (we are measuring diversity at the fitness, genotypic and behavioral level and, differently from the original version we normalised the genotypic diversity). The content of this figure will therefore be updated significantly and we will include an informative caption.
>
> Finally, thank you for the detailed list of typos, we will make sure that our final version has taken them into account.

---

> > ### Comment · Reviewer_FkgN · 2025-11-18
> > **Quick reactions**
> >
> > I really appreciate the author's effort to answer my points quickly to give me an opportunity to provide more feedback. I understand that they would need more time to provide more satisfactory answers.
> >
> > From the authors rebuttal, I believe them may succeed in turning their "rush mode submission" into a decent ICLR paper, given that all reviewers found the topic interesting. I basically agree with their plans for the revised version.
> >
> > About anonimity violation, as another reviewer said, "the anonymous github link in the paper contains references to a file location (under requirements.txt) which reveals the author identity. Also, there is an "online zip" file linked in the README.md file which is a non-anonymised shared zip file, which again shows the author name." From requirements.xt we know the first name of one of the authors, and from there it is easy to discover who they are with a quick search.
> >
> > About not immediately submitting the appendix with the main paper, I also assume this is not intentional, but maybe the conference organizers will need to punish this, so that it does not become a strategy (a simple trick for when your paper is not 100% ready for the deadline)...
> >
> > Good luck for the rest of the rebuttal.

---

> > > ### Author Response · Authors · 2025-11-18
> > > **Thank you for the quick feedback and encouragement**
> > >
> > > Thank you for finding the time to reply and encouraging us to continue with the revision. We will make sure to update you once we have implemented the changes in the previous comments.
> > >
> > > Also thank you for the clarifications on the anonymity issue, a bit late but we will update the repo.
> > >
> > > We understand that uploading the appendix for the first time at the rebuttal is a bad practise and agree with you that it should be discouraged. We confirm that this was not intentional, the current appendix is the one we had prepared (except for the figure we added as response to Comment 2), its rushed nature may support this claim.

---

### Official Review · Reviewer_e6Ea · 2025-10-28

**Soundness:** 2
**Presentation:** 2
**Contribution:** 2
**Rating:** 4
**Confidence:** 2

**Summary:**

Summary
This paper investigates the potential of neuroevolution (NE) as an alternative optimization paradigm for lifelong control, emphasizing its robustness to non-stationary environments where reinforcement learning (RL) often suffers from loss of plasticity. The authors benchmark several NE algorithms—Genetic Algorithm (GA), Evolution Strategies (ES)—against PPO and TRAC-PPO across multiple control domains, including classic control, Minatar, and Kinetix. The study reports that NE methods, especially GA, maintain adaptability under environmental shifts, suggesting that population diversity provides a natural mechanism for continual learning.

**Strengths:**

- **Timely and Conceptually Interesting Topic.**

 The paper tackles a central challenge in continual reinforcement learning—loss of plasticity—and revisits neuroevolution as a biologically inspired, diversity-preserving approach. This perspective is refreshing and well-motivated within the current debate on lifelong learning.

- **Comprehensive Empirical Evaluation.**

 The authors perform a systematic empirical comparison between NE and RL methods across three task families and various network architectures (including a Transformer-based controller), offering an informative dataset on NE’s robustness to environmental shifts.

- **Insightful Qualitative Analysis.**

 The population diversity analysis (Fig. 2–3) provides valuable intuition on how environmental variability induces diversity and triggers phase transitions in evolutionary dynamics, highlighting an underexplored mechanism behind continual adaptation.

**Weaknesses:**

- **Lack of Engineering and Efficiency Analysis.**

 The study does not quantify practical aspects such as wall-clock training time, computational cost, or memory footprint. Matching algorithms only by environment steps ignores the substantial difference in hardware utilization between population-based NE and gradient-based RL. Without these measurements, it is unclear whether NE offers any realistic advantage in deployment scenarios.

- **Limited Task Coverage and Insufficient Hyperparameter Study.**

 The benchmark suite is narrow and mostly low-dimensional. Complex control domains (e.g., DM Control, MetaWorld) and visual tasks are missing. In addition, NE methods use mostly default hyperparameters with minimal tuning; no ablation or sensitivity analysis is provided. As a result, claims of generality and robustness remain preliminary.

- **Insufficient Methodological Formalization.**

 The paper provides only high-level textual descriptions of GA and ES without formal equations, pseudocode, or notation. Readers unfamiliar with NE cannot reconstruct the optimization objectives, mutation/selection mechanisms, or implementation details. The lack of mathematical rigor and algorithmic clarity weakens reproducibility and undermines the technical contribution.

**Questions:**

See the weaknesses above.

---

### Official Review · Reviewer_p5fX · 2025-10-28

**Soundness:** 3
**Presentation:** 3
**Contribution:** 3
**Rating:** 6
**Confidence:** 3

**Summary:**

This paper presents a compelling comparative study between Neuroevolution (NE) and Reinforcement Learning (RL) in the context of lifelong learning, where agents must continuously adapt to environmental shifts. The central thesis is that population-based NE methods inherently possess a superior ability to maintain plasticity compared to gradient-based RL, which often suffers from a "loss of plasticity" in non-stationary environments. The authors benchmark two NE approaches (a Genetic Algorithm and Evolution Strategies) against PPO and a lifelong variant (TRAC-PPO) across three diverse task families: classic control, simplified Atari (Minatar), and a complex physics-based control suite (Kinetix) featuring a Transformer-based policy.  The authors support their claims with an analysis of population diversity, suggesting that environmental shifts naturally promote diversity, which acts as a buffer and facilitates adaptation, sometimes through an abrupt "phase transition" to a better solution.

**Strengths:**

1. The paper focuses on lifelong plasticity, which is a critical issue for embodied AI and robotics.

2. The empirical study is thorough. Using three distinct task families with different network architectures (feedforward, CNN, Transformer) and challenges (sparse rewards, complex dynamics, perceptual shifts) provides a comprehensive evaluation that is rare in the literature. The inclusion of Kinetix, a high-dimensional physics-based environment, is particularly valuable for the robotics community.

3. The consistent outperformance of NE, especially the GA, in the face of environmental shifts is a strong and well-supported result. It effectively challenges the prevailing RL-centric paradigm for continual learning and successfully argues for NE as a powerful, hyperparameter-robust alternative.

**Weaknesses:**

1. The identified weakness of the GA in sparse reward environments (MountainCar, some Minatar games) is a major practical limitation. For robotics, where informative rewards are often hard to engineer, this is a significant drawback. The paper would be stronger if it proposed or discussed potential solutions (e.g., hybridizing NE with novelty search or quality-diversity methods such as MAP-ELITES) to mitigate this.

2. The complete failure of ES on the Transformer-based Kinetix tasks is noted but not sufficiently investigated.

**Questions:**

1. Your results suggest a "best of both worlds" approach. Have you considered or do you plan to investigate hybrid algorithms? such as evolutionary reinforcement learning?

2. Your diversity analysis focuses on genotypic (parameter) diversity. In embodied intelligence, behavioral diversity is often more meaningful. Did you observe a correlation between parameter diversity and behavioral diversity in your populations?

3. In the Minatar and Kinetix experiments, you use a fixed task sequence. Did you observe any evidence of catastrophic forgetting in NE, where performance on a previous task drops after a shift? How does the implicit curriculum induced by your task sequence interact with the evolutionary process?

---

### Official Review · Reviewer_5W23 · 2025-10-29

**Soundness:** 2
**Presentation:** 1
**Contribution:** 3
**Rating:** 2
**Confidence:** 4

**Summary:**

The paper proposes that neuroevolution outperforms traditional reinforcement learning at lifelong learning tasks and provides empirical evidence to back up this claim.
The study compares the genetic algorithm used in Such et al. 2018, the evolution strategy used in Salimans et al. 2017, and PPO, at how well they respond to environmental changes when optimising neural network policies for a suite of control tasks.
The results show that the evolutionary algorithms (EA) do outperform PPO in all of the experiments.
Additionally, an analysis into how the diversity and size of the populations produced by the EAs correlates with performance is carried out.

**Strengths:**

1. The current failure of foundational models to perform continual learning is one of the most pressing current issues in the field of AI, thus the line of inquiry explored in this paper is very important.
2. The findings are interesting and do show that the neuroevolution algorithms outperform PPO at a substantial number of lifelong learning tasks.
3. Multiple experiments were ran for each algorithm and task suggesting robust results.
4. As far as I am aware this is the first study of its kind directly comparing RL and NE algorithms in lifelong learning tasks (although I do not know all the literature in depth).
5. I think it is wise to compare using the number of environment steps as opposed to other metrics, such as wall clock time.

**Weaknesses:**

1. I think more algorithms should have been evaluated in this study in order to provide a more comprehensive comparison. It is unclear whether the relatively poor performance exhibited by PPO is reflective of RL algorithms as a whole or just with this particular algorithm. Also, it seems unusual that a paper studying the plasticity of neuroevolution algorithms does not also include plastic and hebbian neural network algorithms, such as those surveyed in Soltoggio et al 2018. It would have also been interesting to evaluate Novelty Search in order to compare diversity across the behavioural space rather than just the parameter space. Without including these other algorithms the comparison feels incomplete.
2. The paper is written quite carelessly and with many mistakes. There are too many typos to note and many grammatical errors; there are missing references in the bibliography, such as Such et al. 2018; the authors state that both CMAES and TRAC PPO will also be evaluated but these results are not reported; there are missing axes on some of the plots making them unclear; the appendix is referred to but has not been provided; and certain points are repeated multiple times.
3. The abstract claims: 'We observe that, in the presence of environmental shifts, NE naturally increases its diversity of solutions...'; however, the results don't necessarily backup this claim. In fact, the diversity plots for Cartpole in Figure 2 illustrate an acute _decrease_ in diversity at the point that the environment shifts occur, and the other two diversity plots show no increase in diversity with environmental shifts. It is true that Figure 3 shows a marked increase in diversity at generation 9000 correlating with an increase in fitness for the larger population; however, this does seem rather arbitrary considering there have been multiple environmental shifts up until this point and we do not see this jump in those cases. Also, it seems suspicious that all of a sudden the fitness completely stabilises at this arbitrary point in the run despite being highly variable up until this point.
4. It is true that the plots in Figure 3 show a large difference in fitness after the 10000 generation mark between the two population sizes, and that there is also a higher diversity in the larger population; however, this is not necessarily proof that the diversity itself is solely (or at all) responsible for the fitness increase.

**Questions:**

1. Why do you believe a phase transition occurred at generation 9000? Is there anything special about this point? Why do you think the fitness stabilised at this point in particular?
2. Why were more algorithms not compared?

---

### Official Review · Reviewer_CnDj · 2025-11-02

**Soundness:** 2
**Presentation:** 1
**Contribution:** 1
**Rating:** 2
**Confidence:** 4

**Summary:**

The paper proposes NeuroEvolution (NE) as a promising approach for lifelong adaptation due to its ability to increase the diversity of solutions compared to reinforcement learning. Using empirical results, the authors propose NE as an alternative to RL for lifelong learning. The main arguments being that NE is better in terms of plasticity, and that it is better method to maintain diversity.

**Strengths:**

The overall idea is clear, and the authors consider the very important problem of lifelong learning from a non-RL perspective.

**Weaknesses:**

The main issues with the paper is a lack of clarity when it comes to describing the various quantities like diversity, plasticity etc. Due to this, the results are not easy to interpret and the question of what exactly the authors are investigating remains unclear. For instance, it is already known that NE maintains better diversity of solutions/policies (although I am unsure whether this is what the authors refer to when they mention diversity). Apart from this, there is also some ambiguity when it comes to what the main claim is – for instance, it is mentioned that NE is proposed as an alternative to RL. However, it is also mentioned in the discussion that NE is not meant as a replacement to RL, and is meant to be complementary. There are also several typos, grammatical and punctuation errors throughout the manuscript. Some citations are not listed properly (eg: Koza on pg 1). It would be good to clearly summarise all claims at the end of the Introduction. Also, I notice some sections as well as figure references (like fig 6) are present in the appendix. Ideally these should be self contained. With reference to Fig 3, please name the sub figures as (a), (b) etc.,

**Questions:**

What is formally meant by plasticity and diversity?

In line 68, what is meant by “abilities”? What is the context?

NE is designed to be more robust to environmental shifts. So isn’t it expected to perform better?

---

### Author Response · Authors · 2025-11-18
**Plan for revisions based on feedback from all reviewers**

We thank all reviewers for their valuable feedback. We are particularly happy with both their positive reception of the significance of our work and their numerous suggestions for improving the quality of our study. We agree with almost all suggestions and are working on addressing them by the rebuttal deadline. To make the most of the rebuttal, we explain below the things we are working on and are planning to upload. Once we are done with these tasks, we will individually reply to each reviewer. In the meantime, it would be very useful to hear from reviewers in the meantime if they disagree with the proposed additional experiments or think that something is missing.

Upcoming revisions:

1) Report the performance of TRAC-PPO for all tasks in Figure 1

2) Include another RL baseline: DQN with dormant neuron avoidance https://dl.acm.org/doi/10.5555/3618408.3619740

3) Get insights into why RL fails. Here we will establish the loss of plasticity of RL by monitoring different metrics (eg size of gradient updates, behavioral change and number of dormant neurons) as well as forward/backward transfer

4) Investigate evolution with explicit diversity preservation mechanisms. Here we will present Dominated Novelty Search from https://dl.acm.org/doi/pdf/10.1145/3712256.3726310

5) Improve the analysis of the diversity for evolution, presenting diversity at the fitness level (variance across the population), genotypic level (normalised pairwise mean squared distance), and behavioral level (unnormalised pairwise mean squared distance over full state trajectories).  This analysis will replace the current Figure 3

6) Analysis of the effect of frequency of task variation. Here we have results showing that for RL more frequent variations are worse while for evolution the effect is task-dependent and some tasks benefit for more frequent variation.

7) Analysis of robustness to hyperparameters for evolution

8) Evaluate for a task with pixel-based observations (this will be Kinetix, which comes with both the symbolic observations that we have presented and pixel-based).

7) Upload the missing appendices (we already uploaded the version prepared for the original submission but will keep revising)

8) Thorough rewrite of the text to improve presentation

---

### Meta-Review · Area_Chair_j6Pp · 2026-01-07

**Summary:**

Reviewers agree that the paper addresses an important problem (loss of plasticity in lifelong control) and presents an empirical comparison between neuro-evolution and reinforcement learning. However, overall the reviews lean toward rejection, with major concerns about presentation quality, clarity of claims, missing or late supplementary material, limited baseline coverage, and lack of efficiency analysis. While some empirical results are viewed as interesting, reviewers are not fully convinced that the evidence sufficiently supports the paper’s broader conclusions.

**Reviewer Concerns:**

Although an appendix and a detailed revision plan (appreciated by one reviewer) were provided during the rebuttal period, core concerns regarding clarity, baseline completeness, interpretation of diversity results, and practical limitations remain only partially addressed.

**Reviewer Scores:**

Given the mixed, predominantly leaning towards rejection, reviews and the unresolved concerns, reviewer scores were unlikely to change substantially.

---

### Decision · Program_Chairs · 2026-01-26

Reject